# Principal Differences Analysis: Interpretable Characterization of Differences between Distributions

**Jonas Mueller**
CSAIL, MIT
jonasmueller@csail.mit.edu

**Tommi Jaakkola**
CSAIL, MIT
tommi@csail.mit.edu

## Abstract

We introduce principal differences analysis (PDA) for analyzing differences between high-dimensional distributions. The method operates by finding the projection that maximizes the Wasserstein divergence between the resulting univariate populations. Relying on the Cramer-Wold device, it requires no assumptions about the form of the underlying distributions, nor the nature of their inter-class differences. A sparse variant of the method is introduced to identify features responsible for the differences. We provide algorithms for both the original minimax formulation as well as its semidefinite relaxation. In addition to deriving some convergence results, we illustrate how the approach may be applied to identify differences between cell populations in the somatosensory cortex and hippocampus as manifested by single cell RNA-seq. Our broader framework extends beyond the specific choice of Wasserstein divergence.

## 1 Introduction

Understanding differences between populations is a common task across disciplines, from biomedical data analysis to demographic or textual analysis. For example, in biomedical analysis, a set of variables (features) such as genes may be profiled under different conditions (e.g. cell types, disease variants), resulting in two or more populations to compare. The hope of this analysis is to answer whether or not the populations differ and, if so, which variables or relationships contribute most to this difference. In many cases of interest, the comparison may be challenging primarily for three reasons: 1) the number of variables profiled may be large, 2) populations are represented by finite, unpaired, high-dimensional sets of samples, and 3) information may be lacking about the nature of possible differences (exploratory analysis).

We will focus on the comparison of two high dimensional populations. Therefore, given two unpaired i.i.d. sets of samples $\mathbf{X}^{(n)} = x^{(1)}, \ldots, x^{(n)} \sim \mathbb{P}_X$ and $\mathbf{Y}^{(m)} = y^{(1)}, \ldots, y^{(m)} \sim \mathbb{P}_Y$, the goal is to answer the following two questions about the underlying multivariate random variables $X, Y \in \mathbb{R}^d$: (Q1) Is $\mathbb{P}_X = \mathbb{P}_Y$? (Q2) If not, what is the minimal subset of features $S \subseteq \{1, \ldots, d\}$ such that the marginal distributions differ $\mathbb{P}_{X_S} \neq \mathbb{P}_{Y_S}$ while $\mathbb{P}_{X_{S^C}} \approx \mathbb{P}_{Y_{S^C}}$ for the complement? A finer version of (Q2) may additionally be posed which asks how much each feature contributes to the overall difference between the two probability distributions (with respect to the given scale on which the variables are measured).

Many two-sample analyses have focused on characterizing limited differences such as mean shifts [1, 2]. More general differences beyond the mean of each feature remain of interest, however, including variance/covariance of demographic statistics such as income. It is also undesirable to restrict the analysis to specific parametric differences, especially in exploratory analysis where the nature of the underlying distributions may be unknown. In the univariate case, a number of nonparametric tests of equality of distributions are available with accompanying concentration results [3]. Popular examples of such *divergences* (also referred to as probability metrics) include: $f$-divergences

(Kullback-Leibler, Hellinger, total-variation, etc.), the Kolmogorov distance, or the Wasserstein metric [4]. Unfortunately, this simplicity vanishes as the dimensionality $d$ grows, and complex test-statistics have been designed to address some of the difficulties that appear in high-dimensional settings [5, 6, 7, 8].

In this work, we propose the *principal differences analysis* (PDA) framework which circumvents the curse of dimensionality through explicit reduction back to the univariate case. Given a pre-specified statistical divergence $D$ which measures the difference between univariate probability distributions, PDA seeks to find a projection $\beta$ which maximizes $D(\beta^T X, \beta^T Y)$ subject to the constraints $||\beta||_2 \leq 1, \beta_1 \geq 0$ (to avoid underspecification). This reduction is justified by the Cramer-Wold device, which ensures that $\mathbb{P}_X \neq \mathbb{P}_Y$ *if and only if* there exists a direction along which the univariate linearly projected distributions differ [9, 10, 11]. Assuming $D$ is a *positive definite* divergence (meaning it is nonzero between any two distinct univariate distributions), the projection vector produced by PDA can thus capture arbitrary types of differences between high-dimensional $\mathbb{P}_X$ and $\mathbb{P}_Y$. Furthermore, the approach can be straightforwardly modified to address (Q2) by introducing a sparsity penalty on $\beta$ and examining the features with nonzero weight in the resulting optimal projection. The resulting comparison pertains to marginal distributions up to the sparsity level. We refer to this approach as *sparse differences analysis* or SPARDA.

## 2 Related Work

The problem of characterizing differences between populations, including feature selection, has received a great deal of study [2, 12, 13, 5, 1]. We limit our discussion to projection-based methods which, as a family of methods, are closest to our approach. For multivariate two-class data, the most widely adopted methods include (sparse) linear discriminant analysis (LDA) [2] and the logistic lasso [12]. While interpretable, these methods seek specific differences (e.g., covariance-rescaled average differences) or operate under stringent assumptions (e.g., log-linear model). In contrast, SPARDA (with a positive-definite divergence) aims to find features that characterize a priori unspecified differences between general multivariate distributions.

Perhaps most similar to our general approach is Direction-Projection-Permutation (DiProPerm) procedure of Wei et al. [5], in which the data is first projected along the normal to the separating hyperplane (found using linear SVM, distance weighted discrimination, or the centroid method) followed by a univariate two-sample test on the projected data. The projections could also be chosen at random [1]. In contrast to our approach, the choice of the projection in such methods is not optimized for the test statistics. We note that by restricting the divergence measure in our technique, methods such as the (sparse) linear support vector machine [13] could be viewed as special cases. The divergence in this case would measure the margin between projected univariate distributions. While suitable for finding well-separated projected populations, it may fail to uncover more general differences between possibly multi-modal projected populations.

## 3 General Framework for Principal Differences Analysis

For a given divergence measure $D$ between two univariate random variables, we find the projection $\widehat{\beta}$ that solves

$$\max_{\beta \in \mathcal{B}, ||\beta||_0 \leq k} \left\{ D(\beta^T \widehat{X}^{(n)}, \beta^T \widehat{Y}^{(m)}) \right\} \tag{1}$$

where $\mathcal{B} := \{\beta \in \mathbb{R}^d : ||\beta||_2 \leq 1, \beta_1 \geq 0\}$ is the feasible set, $||\beta||_0 \leq k$ is the sparsity constraint, and $\beta^T \widehat{X}^{(n)}$ denotes the observed random variable that follows the empirical distribution of $n$ samples of $\beta^T X$. Instead of imposing a hard cardinality constraint $||\beta||_0 \leq k$, we may instead penalize by adding a penalty term[1] $-\lambda ||\beta||_0$ or its natural relaxation, the $\ell_1$ shrinkage used in Lasso [12], sparse LDA [2], and sparse PCA [14, 15]. Sparsity in our setting explicitly restricts the comparison to the marginal distributions over features with non-zero coefficients. We can evaluate the null hypothesis $\mathbb{P}_X = \mathbb{P}_Y$ (or its sparse variant over marginals) using permutation testing (cf. [5, 16]) with statistic $D(\widehat{\beta}^T \widehat{X}^{(n)}, \widehat{\beta}^T \widehat{Y}^{(m)})$.

The divergence $D$ plays a key role in our analysis. If $D$ is defined in terms of density functions as in $f$-divergence, one can use univariate kernel density estimation to approximate projected pdfs with additional tuning of the bandwidth hyperparameter. For a suitably chosen kernel (e.g. Gaussian), the unregularized PDA objective (without shrinkage) is a smooth function of $\beta$, and thus amenable to the projected gradient method (or its accelerated variants [17, 18]). In contrast, when $D$ is defined over the cdfs along the projected direction – e.g. the Kolmogorov or Wasserstein distance that we focus on in this paper – the objective is nondifferentiable due to the discrete jumps in the empirical cdf. We specifically address the combinatorial problem implied by the Wasserstein distance. Moreover, since the divergence assesses general differences between distributions, Equation (1) is typically a non-concave optimization. To this end, we develop a semi-definite relaxation for use with the Wasserstein distance.

## 4 PDA using the Wasserstein Distance

In the remainder of the paper, we focus on the squared $L_2$ Wasserstein distance (a.k.a. Kantorovich, Mallows, Dudley, or earth-mover distance), defined as

$$D(X, Y) = \min_{\mathbb{P}_{XY}} \mathbb{E}_{\mathbb{P}_{XY}} ||X - Y||^2 \quad \text{s.t.} \quad (X, Y) \sim \mathbb{P}_{XY}, \ X \sim \mathbb{P}_X, \ Y \sim \mathbb{P}_Y \qquad (2)$$

where the minimization is over all joint distributions over $(X, Y)$ with given marginals $\mathbb{P}_X$ and $\mathbb{P}_Y$. Intuitively interpreted as the amount of *work* required to transform one distribution into the other, $D$ provides a natural dissimilarity measure between populations that integrates both the fraction of individuals which are different and the magnitude of these differences. While component analysis based on the Wasserstein distance has been limited to [19], this divergence has been successfully used in many other applications [20]. In the univariate case, (2) may be analytically expressed as the $L_2$ distance between quantile functions. We can thus efficiently compute empirical projected Wasserstein distances by sorting $X$ and $Y$ samples along the projection direction to obtain quantile estimates.

Using the Wasserstein distance, the empirical objective in Equation (1) between unpaired sampled populations $\{x^{(1)}, \dots, x^{(n)}\}$ and $\{y^{(1)}, \dots, y^{(m)}\}$ can be shown to be

$$\max_{\substack{\beta \in \mathcal{B} \\ ||\beta||_0 \leqslant k}} \left\{ \min_{M \in \mathcal{M},} \sum_{i=1}^{n} \sum_{j=1}^{m} (\beta^T x^{(i)} - \beta^T y^{(j)})^2 M_{ij} \right\} = \max_{\substack{\beta \in \mathcal{B} \\ ||\beta||_0 \leqslant k}} \left\{ \min_{M \in \mathcal{M}} \beta^T W_M \beta \right\} \qquad (3)$$

where $\mathcal{M}$ is the set of all $n \times m$ nonnegative *matching* matrices with fixed row sums $= 1/n$ and column sums $= 1/m$ (see [20] for details), $W_M := \sum_{i,j} [Z_{ij} \otimes Z_{ij}] M_{ij}$, and $Z_{ij} := x^{(i)} - y^{(j)}$. If we omitted (fixed) the inner minimization over the matching matrices and set $\lambda = 0$, the solution of (3) would be simply the largest eigenvector of $W_M$. Similarly, for the sparse variant without minizing over $M$, the problem would be solvable as sparse PCA [14, 15, 21]. The actual max-min problem in (3) is more complex and non-concave with respect to $\beta$. We propose a two-step procedure similar to "tighten after relax" framework used to attain minimax-optimal rates in sparse PCA [21]. First, we first solve a convex relaxation of the problem and subsequently run a steepest ascent method (initialized at the global optimum of the relaxation) to greedily improve the current solution with respect to the original nonconvex problem whenever the relaxation is not tight.

Finally, we emphasize that PDA (and SPARDA) not only computationally resembles (sparse) PCA, but the latter is actually a special case of the former in the Gaussian, paired-sample-differences setting. This connection is made explicit by considering the two-class problem with *paired* samples $(x^{(i)}, y^{(i)})$ where $X, Y$ follow two multivariate Gaussian distributions. Here, the largest principal component of the (uncentered) differences $x^{(i)} - y^{(i)}$ is in fact equivalent to the direction which maximizes the projected Wasserstein difference between the distribution of $X - Y$ and a delta distribution at 0.

### 4.1 Semidefinite Relaxation

The SPARDA problem may be expressed in terms of $d \times d$ symmetric matrices $B$ as

$$\max_{B} \min_{M \in \mathcal{M}} \text{tr}(W_M B)$$

$$\text{subject to} \quad \text{tr}(B) = 1, \ B \geq 0, \ ||B||_0 \leqslant k^2, \ \text{rank}(B) = 1 \qquad (4)$$

where the correspondence between (3) and (4) comes from writing $B = \beta \otimes \beta$ (note that any solution of (3) will have unit norm). When $k = d$, i.e., we impose no sparsity constraint as in PDA, we can relax by simply dropping the rank-constraint. The objective is then a supremum of linear functions of $B$ and the resulting semidefinite problem is concave over a convex set and may be written as:

$$\max_{B \in \mathcal{B}_r} \; \min_{M \in \mathcal{M}} \; \mathrm{tr}\left(W_M B\right) \tag{5}$$

where $\mathcal{B}_r$ is the convex set of positive semidefinite $d \times d$ matrices with trace = 1. If $B^* \in \mathbb{R}^{d \times d}$ denotes the global optimum of this relaxation and $\mathrm{rank}(B^*) = 1$, then the best projection for PDA is simply the dominant eigenvector of $B^*$ and the relaxation is tight. Otherwise, we can truncate $B^*$ as in [14], treating the dominant eigenvector as an approximate solution to the original problem (3).

To obtain a relaxation for the sparse version where $k < d$ (SPARDA), we follow [14] closely. Because $B = \beta \otimes \beta$ implies $||B||_0 \leqslant k^2$, we obtain an equivalent cardinality constrained problem by incorporating this nonconvex constraint into (4). Since $\mathrm{tr}(B) = 1$ and $||B||_F = ||\beta||_2^2 = 1$, a convex relaxation of the squared $\ell_0$ constraint is given by $||B||_1 \leqslant k$. By selecting $\lambda$ as the optimal Lagrange multiplier for this $\ell_1$ constraint, we can obtain an equivalent penalized reformulation parameterized by $\lambda$ rather than $k$ [14]. The sparse semidefinite relaxation is thus the following concave problem

$$\max_{B \in \mathcal{B}_r} \left\{ \; \min_{M \in \mathcal{M}} \; \mathrm{tr}\left(W_M B\right) - \lambda ||B||_1 \; \right\} \tag{6}$$

While the relaxation bears strong resemblance to DSPCA relaxation for sparse PCA, the inner maximization over matchings prevents direct application of general semidefinite programming solvers. Let $M(B)$ denote the matching that minimizes $\mathrm{tr}\left(W_M B\right)$ for a given $B$. Standard projected subgradient ascent could be applied to solve (6), where at the $t^{\text{th}}$ iterate the (matrix-valued) subgradient is $W_{M(B^{(t)})}$. However, this approach requires solving optimal transport problems with large $n \times m$ matrices at each iteration. Instead, we turn to a dual form of (6), assuming $n \geqslant m$ (cf. [22, 23])

$$\max_{B \in \mathcal{B}_r, u \in \mathbb{R}^n, v \in \mathbb{R}^m} \frac{1}{m} \sum_{i=1}^{n} \sum_{j=1}^{m} \min\{0, \; \mathrm{tr}([Z_{ij} \otimes Z_{ij}] B) - u_i - v_j\} + \frac{1}{n} \sum_{i=1}^{n} u_i + \frac{1}{m} \sum_{j=1}^{m} v_j - \lambda ||B||_1 \tag{7}$$

(7) is simply a maximization over $B \in \mathcal{B}_r$, $u \in \mathbb{R}^n$, and $v \in \mathbb{R}^m$ which no longer requires matching matrices nor their cumbersome row/column constraints. While dual variables $u$ and $v$ can be solved in closed form for each fixed $B$ (via sorting), we describe a simple sub-gradient approach that works better in practice.

---

**RELAX Algorithm:** Solves the dualized semidefinite relaxation of SPARDA (7). Returns the largest eigenvector of the solution to (6) as the desired projection direction for SPARDA.

---

**Input:** $d$-dimensional data $x^{(1)}, \ldots, x^{(n)}$ and $y^{(1)}, \ldots, y^{(m)}$ (with $n \geqslant m$)
**Parameters:** $\lambda \geqslant 0$ controls the amount of regularization, $\gamma > 0$ is the step-size used for $B$ updates, $\eta > 0$ is the step-size used for updates of dual variables $u$ and $v$, $T$ is the maximum number of iterations without improvement in cost after which algorithm terminates.

1: Initialize $\beta^{(0)} \leftarrow \left[ \frac{\sqrt{d}}{d}, \ldots, \frac{\sqrt{d}}{d} \right]$, $B^{(0)} \leftarrow \beta^{(0)} \otimes \beta^{(0)} \in \mathcal{B}_r$, $u^{(0)} \leftarrow \mathbf{0}_{n \times 1}$, $v^{(0)} \leftarrow \mathbf{0}_{m \times 1}$

2: **While** the number of iterations since last improvement in objective function is less than $T$:

3: $\quad \partial u \leftarrow [1/n, \ldots, 1/n] \in \mathbb{R}^n$, $\partial v \leftarrow [1/m, \ldots, 1/m] \in \mathbb{R}^m$, $\partial B \leftarrow \mathbf{0}_{d \times d}$

4: $\quad$ **For** $i, j \in \{1, \ldots, n\} \times \{1, \ldots, m\}$:

5: $\quad\quad Z_{ij} \leftarrow x^{(i)} - y^{(j)}$

6: $\quad\quad$ **If** $\mathrm{tr}([Z_{ij} \otimes Z_{ij}] B^{(t)}) - u_i^{(t)} - v_j^{(t)} < 0$ :

7: $\quad\quad\quad \partial u_i \leftarrow \partial u_i - 1/m$, $\partial v_j \leftarrow \partial v_j - 1/m$, $\partial B \leftarrow \partial B + Z_{ij} \otimes Z_{ij}/m$

8: $\quad$ **End For**

9: $\quad u^{(t+1)} \leftarrow u^{(t)} + \eta \cdot \partial u$ and $v^{(t+1)} \leftarrow v^{(t)} + \eta \cdot \partial v$

10: $\quad B^{(t+1)} \leftarrow$ **Projection**$\left( B^{(t)} + \frac{\gamma}{||\partial B||_F} \cdot \partial B \; ; \; \lambda, \; \gamma/||\partial B||_F \right)$

**Output:** $\widehat{\beta}_{\text{relax}} \in \mathbb{R}^d$ defined as the largest eigenvector (based on corresponding eigenvalue's magnitude) of the matrix $B^{(t^*)}$ which attained the best objective value over all iterations.

---

**Projection Algorithm:** Projects matrix onto positive semidefinite cone of unit-trace matrices $\mathcal{B}_r$ (the feasible set in our relaxation). Step 4 applies soft-thresholding proximal operator for sparsity.

---

**Input:** $B \in \mathbb{R}^{d \times d}$
**Parameters:** $\lambda \geqslant 0$ controls the amount of regularization, $\delta = \gamma / ||\partial B||_F \geqslant 0$ is the actual step-size used in the $B$-update.

1: $Q \Lambda Q^T \leftarrow$ eigendecomposition of $B$

2: $w^* \leftarrow \arg\min \left\{ ||w - \mathrm{diag}(\Lambda)||_2^2 : w \in [0,1]^d, ||w||_1 = 1 \right\}$         (Quadratic program)

3: $\widetilde{B} \leftarrow Q \cdot \mathrm{diag}\{w_1^*, \ldots, w_d^*\} \cdot Q^T$

4: **If** $\lambda > 0$: **For** $r, s \in \{1, \ldots, d\}^2$:      $\widetilde{B}_{r,s} \leftarrow \mathrm{sign}(\widetilde{B}_{r,s}) \cdot \max\{0, |\widetilde{B}_{r,s}| - \delta\lambda\}$
**Output:** $\widetilde{B} \in \mathcal{B}_r$

---

The RELAX algorithm (boxed) is a projected subgradient method with supergradients computed in Steps 3 - 8. For scaling to large samples, one may alternatively employ *incremental* supergradient directions [24] where Step 4 would be replaced by drawing random $(i, j)$ pairs. After each subgradient step, projection back into the feasible set $\mathcal{B}_r$ is done via a quadratic program involving the current solution's eigenvalues. In SPARDA, sparsity is encouraged via the soft-thresholding proximal map corresponding to the $\ell_1$ penalty. The overall form of our iterations matches subgradient-proximal updates (4.14)-(4.15) in [24]. By the convergence analysis in §4.2 of [24], the RELAX algorithm (as well as its incremental variant) is guaranteed to approach the optimal solution of the dual which also solves (6), provided we employ sufficiently large $T$ and small step-sizes. In practice, fast and accurate convergence is attained by: (a) renormalizing the $B$-subgradient (Step 10) to ensure balanced updates of the unit-norm constrained $B$, (b) using diminishing learning rates which are initially set larger for the unconstrained dual variables (or even taking multiple subgradient steps in the dual variables per each update of $B$).

## 4.2 Tightening after relaxation

It is unreasonable to expect that our semidefinite relaxation is always tight. Therefore, we can sometimes further refine the projection $\widehat{\beta}_{\mathrm{relax}}$ obtained by the RELAX algorithm by using it as a starting point in the original non-convex optimization. We introduce a sparsity constrained *tightening* procedure for applying projected gradient ascent for the original nonconvex objective $J(\beta) = \min_{M \in \mathcal{M}} \beta^T W_M \beta$ where $\beta$ is now forced to lie in $\mathcal{B} \cap \mathcal{S}_k$ and $\mathcal{S}_k := \{\beta \in \mathbb{R}^d : ||\beta||_0 \leqslant k\}$. The sparsity level $k$ is fixed based on the relaxed solution ($k = ||\widehat{\beta}_{\mathrm{relax}}||_0$). After initializing $\beta^{(0)} = \widehat{\beta}_{\mathrm{relax}} \in \mathbb{R}^d$, the tightening procedure iterates steps in the gradient direction of $J$ followed by straightforward projections into the unit half-ball $\mathcal{B}$ and the set $\mathcal{S}_k$ (accomplished by greedily truncating all entries of $\beta$ to zero besides the largest $k$ in magnitude).

Let $M(\beta)$ again denote the matching matrix chosen in response to $\beta$. $J$ fails to be differentiable at the $\widetilde{\beta}$ where $M(\widetilde{\beta})$ is not unique. This occurs, e.g., if two samples have identical projections under $\widetilde{\beta}$. While this situation becomes increasingly likely as $n, m \to \infty$, $J$ interestingly becomes smoother overall (assuming the distributions admit density functions). For all other $\beta$: $M(\beta') = M(\beta)$ where $\beta'$ lies in a small neighborhood around $\beta$ and $J$ admits a well-defined gradient $2 W_{M(\beta)} \beta$. In practice, we find that the tightening always approaches a local optimum of $J$ with a diminishing stepsize. We note that, for a given projection, we can efficiently calculate gradients without recourse to matrices $M(\beta)$ or $W_{M(\beta)}$ by sorting $\beta^{(t)T} x^{(1)}, \ldots, \beta^{(t)T} x^{(n)}$ and $\beta^{(t)T} y^{(1)}, \ldots, \beta^{(t)T} y^{(m)}$. The gradient is directly derivable from expression (3) where the nonzero $M_{ij}$ are determined by appropriately matching empirical quantiles (represented by sorted indices) since the univariate Wasserstein distance is simply the $L_2$ distance between quantile functions [20]. Additional computation can be saved by employing insertion sort which runs in nearly linear time for almost sorted points (in iteration $t - 1$, the points have been sorted along the $\beta^{(t-1)}$ direction and their sorting in direction $\beta^{(t)}$ is likely similar under small step-size). Thus the tightening procedure is much more efficient than the RELAX algorithm (respective runtimes are $O(dn \log n)$ vs. $O(d^3 n^2)$ per iteration).

We require the combined steps for good performance. The projection found by the tightening algorithm heavily depends on the starting point $\beta^{(0)}$, finding only the closest local optimum (as in Figure 1a). It is thus important that $\beta^{(0)}$ is already a good solution, as can be produced by our RELAX algorithm. Additionally, we note that as first-order methods, both the RELAX and tightening algorithms are amendable to a number of (sub)gradient-acceleration schemes (e.g. momentum techniques, adaptive learning rates, or FISTA and other variants of Nesterov's method [18, 17, 25]).

### 4.3 Properties of semidefinite relaxation

We conclude the algorithmic discussion by highlighting basic conditions under which our PDA relaxation is tight. Assuming $n, m \to \infty$, each of (i)-(iii) implies that the $B^*$ which maximizes (5) is nearly rank one, or equivalently $B^* \approx \widetilde{\beta} \otimes \widetilde{\beta}$ (see Supplementary Information §S4 for intuition). Thus, the tightening procedure initialized at $\widetilde{\beta}$ will produce a global maximum of the PDA objective.

(i) There exists direction in which the *projected* Wasserstein distance between $X$ and $Y$ is nearly as large as the overall Wasserstein distance in $\mathbb{R}^d$. This occurs for example if $||\mathbb{E}[X] - \mathbb{E}[Y]||_2$ is large while both $||\mathrm{Cov}(X)||_F$ and $||\mathrm{Cov}(Y)||_F$ are small (the distributions need not be Gaussian).

(ii) $X \sim N(\mu_X, \Sigma_X)$ and $Y \sim N(\mu_Y, \Sigma_Y)$ with $\mu_X \neq \mu_Y$ and $\Sigma_X \approx \Sigma_Y$.

(iii) $X \sim N(\mu_X, \Sigma_X)$ and $Y \sim N(\mu_Y, \Sigma_Y)$ with $\mu_X = \mu_Y$ where the underlying covariance structure is such that $\arg\max_{B \in \mathcal{B}_r} ||(B^{1/2}\Sigma_X B^{1/2})^{1/2} - (B^{1/2}\Sigma_Y B^{1/2})^{1/2}||_F^2$ is nearly rank 1. For example, if the primary difference between covariances is a shift in the marginal variance of some features, i.e. $\Sigma_Y \approx V \cdot \Sigma_X$ where $V$ is a diagonal matrix.

## 5 Theoretical Results

In this section, we characterize statistical properties of an empirical divergence-maximizing projection $\widehat{\beta} := \arg\max_{\beta \in \mathcal{B}} D(\beta^T \widehat{X}^{(n)}, \beta^T \widehat{Y}^{(n)})$, although we note that the algorithms may not succeed in finding such a global maximum for severely nonconvex problems. Throughout, $D$ denotes the squared $L_2$ Wasserstein distance between univariate distributions, $C$ represents universal constants that change from line to line. All proofs are relegated to the Supplementary Information §S3. We make the following simplifying assumptions: (A1) $n = m$ (A2) $X, Y$ admit continuous density functions (A3) $X, Y$ are compactly supported with nonzero density in the Euclidean ball of radius $R$. Our theory can be generalized beyond (A1)-(A3) to obtain similar (but complex) statements through careful treatment of the distributions' tails and zero-density regions where cdfs are flat.

**Theorem 1.** *Suppose there exists direction $\beta^* \in \mathcal{B}$ such that $D(\beta^{*T}X, \beta^{*T}Y) \geqslant \Delta$. Then:*

$$D(\widehat{\beta}^T \widehat{X}^{(n)}, \widehat{\beta}^T \widehat{Y}^{(n)}) > \Delta - \epsilon \quad \text{with probability greater than } 1 - 4\exp\left(-\frac{n\epsilon^2}{16R^4}\right)$$

Theorem 1 gives basic concentration results for the projections used in empirical applications our method. To relate distributional differences between $X, Y$ in the ambient $d$-dimensional space with their estimated divergence along the univariate linear representation chosen by PDA, we turn to Theorems 2 and 3. Finally, Theorem 4 provides sparsistency guarantees for SPARDA in the case where $X, Y$ exhibit large differences over a certain feature subset (of known cardinality).

**Theorem 2.** *If $X$ and $Y$ are identically distributed in $\mathbb{R}^d$, then: $D(\widehat{\beta}^T \widehat{X}^{(n)}, \widehat{\beta}^T \widehat{Y}^{(n)}) < \epsilon$ with probability greater than*

$$1 - C_1 \left(1 + \frac{R^2}{\epsilon}\right)^d \exp\left(-\frac{C_2}{R^4}n\epsilon^2\right)$$

To measure the difference between the untransformed random variables $X, Y \in \mathbb{R}^d$, we define the following metric between distributions on $\mathbb{R}^d$ which is parameterized by $a \geqslant 0$ (cf. [11]):

$$T_a(X, Y) := |\Pr(|X_1| \leqslant a, \ldots, |X_d| \leqslant a) - \Pr(|Y_1| \leqslant a, \ldots, |Y_d| \leqslant a)| \tag{8}$$

In addition to (A1)-(A3), we assume the following for the next two theorems: (A4) $Y$ has sub-Gaussian tails, meaning cdf $F_Y$ satisfies: $1 - F_Y(y) \leqslant \frac{C}{y}\exp(-y^2/2)$, (A5) $\mathbb{E}[X] = \mathbb{E}[Y] = 0$ (note that mean differences can trivially be captured by linear projections, so these are not the differences of interest in the following theorems), (A6) $\mathrm{Var}(X_\ell) = 1$ for $\ell = 1, \ldots, d$

**Theorem 3.** *Suppose $\exists\, a \geqslant 0$ s.t. $T_a(X, Y) > h\,(g(\Delta))$ where $h\,(g(\Delta)) := \min\{\Delta_1, \Delta_2\}$ with*

$$\Delta_1 := (a+d)^d(g(\Delta)+d) + \exp(-a^2/2) + \psi\exp\left(-1/(\sqrt{2}\psi)\right) \tag{9}$$

$$\Delta_2 := \left(g(\Delta) + \exp(-a^2/2)\right)\cdot d \tag{10}$$

$\psi := ||Cov(X)||_1,\ g(\Delta) := \Delta^4\cdot(1+\Phi)^{-4},\ and\ \Phi := \sup_{\alpha\in\mathcal{B}}\left\{\sup_y|f_{\alpha^T Y}(y)|\right\}$
*with $f_{\alpha^T Y}(y)$ defined as the density of the projection of $Y$ in the $\alpha$ direction.*
*Then:*

$$D(\widehat{\beta}^T\widehat{X}^{(n)}, \widehat{\beta}^T\widehat{Y}^{(n)}) > C\Delta - \epsilon \tag{11}$$

*with probability greater than $1 - C_1\exp\left(-\frac{C_2}{R^4}n\epsilon^2\right)$*

**Theorem 4.** *Define $C$ as in (11). Suppose there exists feature subset $S \subset \{1, \ldots, d\}$ s.t. $|S| = k$, $T(X_S, Y_S) \geqslant h\,(g\,(\epsilon(d+1)/C))$, and remaining marginal distributions $X_{S^C}$, $Y_{S^C}$ are identical. Then:*

$$\widehat{\beta}^{(k)} := \arg\max_{\beta\in\mathcal{B}}\{D(\beta^T\widehat{X}^{(n)}, \beta^T\widehat{Y}^{(n)}) : ||\beta||_0 \leqslant k\}$$

*satisfies $\widehat{\beta}_i^{(k)} \neq 0$ and $\widehat{\beta}_j^{(k)} = 0\ \ \forall\, i \in S, j \in S^C$ with probability greater than*

$$1 - C_1\left(1 + \frac{R^2}{\epsilon}\right)^{d-k}\exp\left(-\frac{C_2}{R^4}n\epsilon^2\right)$$

## 6 Experiments

Figure 1a illustrates the cost function of PDA pertaining to two 3-dimensional distributions (see details in Supplementary Information §S1). In this example, the point of convergence $\widehat{\beta}$ of the tightening method after random initialization (in green) is significantly inferior to the solution produced by the RELAX algorithm (in red). It is therefore important to use RELAX before tightening as we advise.

The synthetic MADELON dataset used in the NIPS 2003 feature selection challenge consists of points ($n = m = 1000, d = 500$) which have 5 features scattered on the vertices of a five-dimensional hypercube (so that interactions between features must be considered in order to distinguish the two classes), 15 features that are noisy linear combinations of the original five, and 480 useless features [26]. While the focus of the challenge was on extracting features useful to classifiers, we direct our attention toward more interpretable models. Figure 1b demonstrates how well SPARDA (red), the top sparse principal component (black) [27], sparse LDA (green) [2], and the logistic lasso (blue) [12] are able to identify the 20 relevant features over different settings of their respective regularization parameters (which determine the cardinality of the vector returned by each method). The red asterisk indicates the SPARDA result with $\lambda$ automatically selected via our cross-validation procedure (without information of the underlying features' importance), and the black asterisk indicates the best reported result in the challenge [26].

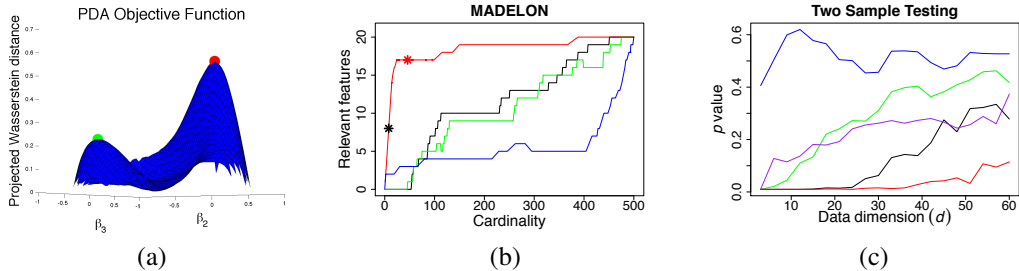

(a)              (b)              (c)

Figure 1: (a) example where PDA is nonconvex, (b) SPARDA vs. other feature selection methods, (c) power of various tests for multi-dimensional problems with 3-dimensional differences.

The restrictive assumptions in logistic regression and linear discriminant analysis are not satisfied in this complex dataset resulting in poor performance. Despite being class-agnostic, PCA was successfully utilized by numerous challenge participants [26], and we find that the sparse PCA performs on par with logistic regression and LDA. Although the lasso fairly efficiently picks out 5 relevant features, it struggles to identify the rest due to severe multi-colinearity. Similarly, the challenge-winning Bayesian SVM with Automatic Relevance Determination [26] only selects 8 of the 20 relevant features. In many applications, the goal is to thoroughly characterize the set of differences rather than select a subset of features that maintains predictive accuracy. SPARDA is better suited for this alternative objective. Many settings of $\lambda$ return 14 of the relevant features with zero false positives. If $\lambda$ is chosen automatically through cross-validation, the projection returned by SPARDA contains 46 nonzero elements of which 17 correspond to relevant features.

Figure 1c depicts (average) $p$-values produced by SPARDA (red), PDA (purple), the overall Wasserstein distance in $\mathbb{R}^d$ (black), Maximum Mean Discrepancy [8] (green), and DiProPerm [5] (blue) in two-sample synthetically controlled problems where $\mathbb{P}_X \neq \mathbb{P}_Y$ and the underlying differences have varying degrees of sparsity. Here, $d$ indicates the overall number of features included of which only the first 3 are relevant (see Supplementary Information §S1 for details). As we evaluate the significance of each method's statistic via permutation testing, all the tests are guaranteed to exactly control Type I error [16], and we thus only compare their respective power in determining $\mathbb{P}_X \neq \mathbb{P}_Y$ setting. The figure demonstrates clear superiority of SPARDA which leverages the underlying sparsity to maintain high power even with the increasing overall dimensionality. Even when all the features differ (when $d = 3$), SPARDA matches the power of methods that consider the full space despite only selecting a single direction (which cannot be based on mean-differences as there are none in this controlled data). This experiment also demonstrate that the unregularized PDA retains greater power than DiProPerm, a similar projection-based method [5].

Recent technological advances allow complete transcriptome profiling in thousands of individual cells with the goal of fine molecular characterization of cell populations (beyond the crude average-tissue-level expression measure that is currently standard) [28]. We apply SPARDA to expression measurements of 10,305 genes profiled in 1,691 single cells from the somatosensory cortex and 1,314 hippocampus cells sampled from the brains of juvenile mice [29]. The resulting $\widehat{\beta}$ identifies many previously characterized subtype-specific genes and is in many respects more informative than the results of standard differential expression methods (see Supplementary Information §S2 for details). Finally, we also apply SPARDA to normalized data with mean-zero & unit-variance marginals in order to explicitly restrict our search to genes whose relationship with other genes' expression is different between hippocampus and cortex cells. This analysis reveals many genes known to be heavily involved in signaling, regulating important processes, and other forms of functional interaction between genes (see Supplementary Information §S2.1 for details). These types of important changes cannot be detected by standard differential expression analyses which consider each gene in isolation or require gene-sets to be explicitly identified as features [28].

## 7 Conclusion

This paper introduces the overall principal differences methodology and demonstrates its numerous practical benefits of this approach. While we focused on algorithms for PDA & SPARDA tailored to the Wasserstein distance, different divergences may be better suited for certain applications.

Further theoretical investigation of the SPARDA framework is of interest, particularly in the high-dimensional $d = O(n)$ setting. Here, rich theory has been derived for compressed sensing and sparse PCA by leveraging ideas such as restricted isometry or spiked covariance [15]. A natural question is then which analogous properties of $\mathbb{P}_X, \mathbb{P}_Y$ theoretically guarantee the strong empirical performance of SPARDA observed in our high-dimensional applications. Finally, we also envision extensions of the methods presented here which employ multiple projections in succession, or adapt the approach to non-pairwise comparison of multiple populations.

**Acknowledgements**

This research was supported by NIH Grant T32HG004947.

## Footnotes

[1] In practice, shrinkage parameter $\lambda$ (or explicit cardinality constraint $k$) may be chosen via cross-validation by maximizing the divergence between held-out samples.

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
