[Supplementary Material]

# Supplementary Information

## Contents

## List of Figures

## List of Tables

## 1 Details of simulation study

To generate the cost function depicted in Figure 1a, we draw $n = m = 1000$ points from mean-zero 3-dimensional Gaussian distributions with the following respective covariance matrices:

$$\Sigma_X = \begin{bmatrix} 1 & 0.2 & 0.4 \\ 0.2 & 1 & 0 \\ 0.4 & 0 & 1 \end{bmatrix} \qquad \Sigma_Y = \begin{bmatrix} 1 & -0.9 & 0 \\ -0.9 & 1 & 0 \\ 0 & 0 & 1 \end{bmatrix}$$

Due to the large sample sizes, the empirical distributions accurately represent the underlying populations, and thus the projection produced by the tightening procedure (in green) is significantly inferior to the projection produced by the RELAX algorithm (in red) in terms of actual divergence captured. Note that only dimensions 2 and 3 of the projection-space are plotted in the figure since $\beta_1 = \sqrt{1 - \sum_{\ell=2}^{d} \beta_\ell^2}$ is fixed for the unit-norm projections of interest.

Next, we detail the process by which the data are generated for the two-sample problems depicted in Figure 1c. We set the features of the underlying $X$ and $Y$ to mean-zero multivariate Gaussian distributions in blocks of 3, where within each block, (common) covariance parameters are sampled from the Wishart($\mathbf{I}_{3 \times 3}$) distribution with 3 degrees of freedom. Only for the first block of 3 features do we sample a separate covariance matrix for $X$ and a separate covariance matrix for $Y$, so all differences between the two distributions lie in the first 3 features. To generate a dataset with $d = 3 \times \ell$, we simply concatenate $\ell$ of our blocks together (always including the first block with the

underlying difference) and draw $n = m = 100$ points from each class. We generate 20 datasets by increasing $\ell$ (so the largest $d = 60$), and repeat this entire experiment 10 times reporting the average $p$-values in Figure 1c.

Each $p$-value is obtain by randomly permuting the class labels and recomputing the test statistic 100 times (where we use the same permutations between all datasets). In SPARDA, regularization parameter $\lambda$ is re-selected using our cross-validation technique in each permutation. The overall Wasserstein distance in the ambient space is computed by solving a transportation problem [1], and we note the similarity between this statistic and the cross-match test [2]. A popular kernel method for testing high-dimensional distribution equality, the mean map discrepancy, is computed using the Gaussian kernel with bandwidth parameter chosen by the "median trick" [3] (this is very similar to the energy test of [4]). Finally, we also compute the DiProPerm statistic, employing the the DWD-$t$ variant recommended for testing general equality of distributions [5].

## 2   Single cell gene expression in cortex vs. hippocampus

Playing critical roles in the brain, the somatosensory cortex (linked to the senses) and hippocampal region (linked to memory regulation and spatial coding) contain a diversity of cell types [6]. It is thus of great interest to identify how cell populations in these regions diverge in developing brains, a question we address by applying SPARDA to single cell RNA-seq data from these regions. Following [7], we represent gene expression by log-transformed FPKM computed from the sequencing read counts[1], so values are directly comparable between genes. Because expression measurements from individual cells are poorer in quality than transcriptome profiles obtained in aggregate across tissue samples (due to a drastically reduced amount of available RNA), it is important to filter out poorly measured genes and we retain a set of 10,305 genes that are measured with sufficient accuracy for informative analysis [7].

Table 1 and Figure 1 demonstrate that SPARDA discovers many interesting genes which are already known to play important functional roles in these regions of the brain. For comparison, we also run LIMMA, a standard method for differential expression analysis which tests for marginal mean-differences on a gene-by-gene basis [8]. Ordering the significant genes under LIMMA by magnitude of their mean expression difference, we find that 3 of the top 10 genes identified by SPARDA also appear in this top 10 list (*Crym*, *Spink8*, *Neurod6*), demonstrating SPARDA's implicit attraction toward large first-order differences over more nuanced changes in practice. Because only few genes can feasibly be considered for subsequent experimentation in these studies, a good tool for differential expression analysis must rank the most relevant genes very highly in order for researchers to take note.

One particularly relevant gene in this data is *Snca*, a presynaptic signaling and membrane trafficking gene whose defects are implicated in both Parkinson and Alzheimer's disease [9, 10]. While *Snca* is ranked 11th highest under SPARDA, it only ranks 349 according to LIMMA $p$-values and 95 based on absolute mean-shift. Figure 2 shows that the primary change in *Snca* expression between the cell types is not a shift in the distributions, but rather the movement of a large fraction of low (1-2.5 log-FPKM) expression cells into the high-expression ($> 2.5$ log-FPKM) regime. As this type of change does not match the restrictive assumptions of LIMMA's $t$-test, the method fails to highly-rank this gene while the Wasserstein distance employed by SPARDA is perfectly suited for measuring this sort of effect.

### 2.1   Identifying genes whose interactions differ between cortex vs. hippocampus cells

After restricting our analysis to only the top 500 genes with largest average expression (since genes playing important roles in interactions must be highly expressed), we normalize each gene's expression values to have mean zero and unit variance within in the cells of each class. Subsequent application of SPARDA reveals that most of the genes corresponding to the ten greatest values of the resulting $\hat{\beta}$ are known to play important roles in in signaling and regulation (see Table 2).

| GENE | WEIGHT | DESCRIPTION |
|---|---|---|
| Cck | 0.0593 | Primary distinguishing gene between distinct interneuron classes identified in the cortex and hippocampus [11] |
| Neurod6 | 0.0583 | General regulator of nervous system development whose induced mutation displays different effects in neocortex vs. the hippocampal region [12] |
| Stmn3 | 0.0573 | Up-expressed in hippocampus of patients with depressive disorders [13] |
| Plp1 | 0.0570 | An oligodendrocyte- and myelin-related gene which exhibits cortical differential expression in schizophrenia [14] |
| Crym | 0.0550 | Plays a role in neuronal specification [15] |
| Spink8 | 0.0536 | Serine protease inhibitor specific to hippocampal pyramidal cells [6] |
| Gap43 | 0.0511 | Encodes plasticity protein important for axonal regeneration and neural growth |
| Cryab | 0.0500 | Stress induction leads to reduced expression in the mouse hippocampus [16] |
| Mal | 0.0494 | Regulates dendritic morphology and is expressed at lower levels in cortex than in hippocampus [17] |
| Tspan13 | 0.0488 | Membrane protein which mediates signal transduction events in cell development, activation, growth and motility |

Table 1: Genes with the greatest weight in the projection $\widehat{\beta}$ produced by SPARDA analysis of the mouse brain single cell RNA-seq data. Where not cited, the description of the genes are taken from the standard ontology annotations.

| GENE | WEIGHT | DESCRIPTION |
|---|---|---|
| Thy1 | 0.1245 | Plays a role in cell-cell & cell-ligand interactions during synaptogenesis and other processes in the brain |
| Vsnl1 | 0.1245 | Modulates intracellular signaling pathways of the central nervous system |
| Stmn3 | 0.1222 | Stathmins form important protein complex with tubulins |
| Stmn2 | 0.1188 | Note: Tubulins Tubb3 and Tubb2 are ranked $20^{\text{th}}$ and $25^{\text{th}}$ by weight in $\widehat{\beta}$ |
| Tmem59 | 0.1176 | Fundamental regulator of neural cell differentiation. Knock out in the hippocampus results in drastic expression changes of many other genes [18] |
| Basp1 | 0.1171 | Transcriptional cofactor which can divert the differentiation of cells to a neuronal-like morphology [19] |
| Snhg1 | 0.1166 | Unclassified non-coding RNA gene |
| Mllt11 | 0.1145 | Promoter of neurodifferentiation and axonal/dendritic maintenance [20] |
| Uchl1 | 0.1137 | Loss of function leads to profound degeneration of motor neurons [21]. |
| Cck | 0.1131 | Targets pyramidal neurons and enables neocortical plasticity allowing for example the auditory cortex to detect light stimuli [22, 23] |

Table 2: Genes with the greatest weight in the projection $\widehat{\beta}$ produced by SPARDA analysis of the marginally normalized expression data.

| gene ontology term | | category, level | set size | candidates contained | p-value | q-value |
|---|---|---|---|---|---|---|
| GO:0019226 | transmission of nerve impulse | BP 4 | 490 | 18 (3.7%) | 5.46e-11 | 1.31e-08 |
| GO:0007268 | synaptic transmission | BP 4 | 391 | 13 (3.3%) | 1.04e-07 | 1.02e-05 |
| GO:0055082 | cellular chemical homeostasis | BP 4 | 632 | 16 (2.5%) | 1.27e-07 | 1.02e-05 |
| GO:0032051 | clathrin light chain binding | MF 4 | 3 | 3 (100.0%) | 1.33e-07 | 4.66e-06 |
| GO:0048666 | neuron development | BP 4 | 646 | 16 (2.5%) | 1.87e-07 | 1.09e-05 |
| GO:0022008 | neurogenesis | BP 4 | 1029 | 20 (1.9%) | 2.28e-07 | 1.09e-05 |
| GO:0032846 | positive regulation of homeostatic process | BP 4 | 57 | 6 (10.5%) | 4.72e-07 | 1.89e-05 |
| GO:0048878 | chemical homeostasis | BP 4 | 838 | 17 (2.0%) | 1.12e-06 | 3.82e-05 |
| GO:0007399 | nervous system development | BP 4 | 1486 | 23 (1.6%) | 1.31e-06 | 3.93e-05 |
| GO:0030182 | neuron differentiation | BP 4 | 854 | 17 (2.0%) | 1.57e-06 | 4.18e-05 |
| GO:0031175 | neuron projection development | BP 4 | 529 | 13 (2.5%) | 3.21e-06 | 7.7e-05 |
| GO:0051969 | regulation of transmission of nerve impulse | BP 4 | 194 | 8 (4.1%) | 7.32e-06 | 0.00016 |
| GO:0048858 | cell projection morphogenesis | BP 4 | 516 | 12 (2.3%) | 1.37e-05 | 0.000275 |
| GO:0032990 | cell part morphogenesis | BP 4 | 542 | 12 (2.2%) | 2.19e-05 | 0.000405 |
| GO:0007010 | cytoskeleton organization | BP 4 | 763 | 14 (1.8%) | 3.33e-05 | 0.000571 |
| GO:0048168 | regulation of neuronal synaptic plasticity | BP 4 | 38 | 4 (10.5%) | 4.29e-05 | 0.000686 |
| GO:0000902 | cell morphogenesis | BP 4 | 814 | 14 (1.7%) | 6.91e-05 | 0.00093 |
| GO:0050877 | neurological system process | BP 4 | 2024 | 24 (1.2%) | 6.97e-05 | 0.00093 |
| GO:0044057 | regulation of system process | BP 4 | 427 | 10 (2.3%) | 7.09e-05 | 0.00093 |
| GO:0008366 | axon ensheathment | BP 4 | 84 | 5 (6.0%) | 7.36e-05 | 0.00093 |
| GO:0008344 | adult locomotory behavior | BP 4 | 86 | 5 (5.8%) | 8.23e-05 | 0.000988 |
| GO:0007611 | learning or memory | BP 4 | 151 | 6 (4.0%) | 0.000131 | 0.0015 |
| GO:0006900 | membrane budding | BP 4 | 21 | 3 (14.3%) | 0.000165 | 0.0018 |
| GO:0071822 | protein complex subunit organization | BP 4 | 900 | 14 (1.6%) | 0.000192 | 0.00201 |
| GO:0001662 | behavioral fear response | BP 4 | 27 | 3 (11.1%) | 0.000356 | 0.00341 |
| GO:0002209 | behavioral defense response | BP 4 | 27 | 3 (11.1%) | 0.000356 | 0.00341 |
| GO:0030913 | paranodal junction assembly | BP 4 | 6 | 2 (33.3%) | 0.00039 | 0.0036 |
| GO:0007626 | locomotory behavior | BP 4 | 188 | 6 (3.2%) | 0.000405 | 0.0036 |

Figure 1: Biological process terms most significantly enriched in the annotations of the top 100 genes identified by SPARDA.

Figure 2: Distribution of *Snca* expression across cells of the somatosensory cortex and hippocampus.

# 3 Proofs and Auxiliary Lemmas

Throughout this section, we use $C$ to denote absolute constants whose value may change from line to line. $F$ is defined the cdf of a random variable, and the corresponding quantile function is $F^{-1}(p) := \inf\{x : F(x) \geqslant p\}$. Note our assumptions (A1)-(A3) ensure the quantile function equals the unique inverse of any projected cdf. Hat notation is used to represent the empirical versions of all quantities, and recall that $D$ denotes the *squared* Wasserstein distance.

## 3.1 Proof of Theorem 1

*Proof.* Since $\widehat{\beta}$ maximizes the empirical divergence, we have:

$$\Pr(D(\widehat{\beta}^T \widehat{X}^{(n)}, \widehat{\beta}^T \widehat{Y}^{(n)}) > \Delta - \epsilon)$$

$$\geqslant \Pr(D(\beta^{*T} \widehat{X}^{(n)}, \beta^{*T} \widehat{Y}^{(n)}) > \Delta - \epsilon)$$

$$\geqslant \Pr(D(\beta^{*T} \widehat{X}^{(n)}, \beta^{*T} X) + D(\beta^{*T} \widehat{Y}^{(n)}, \beta^{*T} Y) \leqslant \epsilon)$$

$$\geqslant 1 - 4 \exp\left(-\frac{n\epsilon^2}{16R^4}\right) \quad \text{applying Lemma 1 and the union bound.}$$

$\square$

**Lemma 1.** *For bounded univariate random variable $Z \in [-R, R]$ with nonzero continuous density in this region, we have*

$$D(\widehat{Z}^{(n)}, Z) > \epsilon$$

*with probability at most* $2 \exp\left(-\frac{n\epsilon^2}{8R^4}\right)$

*Proof.* On the real line, the (squared) Wasserstein distance is given by:

$$D(\widehat{Z}^{(n)}, Z) = \int_0^1 \left(\widehat{F}_Z^{-1}(p) - F_Z^{-1}(p)\right)^2 dp$$

$$= 4R^2 \int_0^1 \left(\frac{\widehat{F}_Z^{-1}(p) - F_Z^{-1}(p)}{2R}\right)^2 dp \quad \text{where} \quad \left|\frac{\widehat{F}_Z^{-1}(p) - F_Z^{-1}(p)}{2R}\right| \leqslant 1 \text{ for each } p \in (0, 1)$$

$$\leqslant 4R^2 \int_0^1 \left|\frac{\widehat{F}_Z^{-1}(p) - F_Z^{-1}(p)}{2R}\right| dp$$

$$= 2R \int_0^1 \left|\widehat{F}_Z^{-1}(p) - F_Z^{-1}(p)\right| dp$$

$$= 2R \int_{-\infty}^{\infty} \left|\widehat{F}_Z(z) - F_Z(z)\right| dz \quad \text{by the equivalence of the (empirical) quantile function and inverse (empirical) cdf}$$

$$\leqslant 4R^2 \cdot \sup_z \left|\widehat{F}_Z(z) - F_Z(z)\right|$$

$$\leqslant \epsilon \text{ with probability } \geqslant 1 - 2 \exp\left(-\frac{n\epsilon^2}{8R^4}\right) \quad \text{by the Dvoretzky-Kiefer-Wolfowitz inequality [24].}$$

$\square$

## 3.2 Proof of Theorem 2

*Proof.* We first construct a fine grid of points $\{\alpha_1, \dots, \alpha_S\}$ which form an $(\epsilon/R^2)$-net covering the surface of the unit ball in $\mathbb{R}^d$. When $\mathbb{P}_X = \mathbb{P}_Y$, the Cramer-Wold device [25] implies that for any point in our grid: $D(\alpha_s^T X, \alpha_s^T Y) = 0$. A result analogous to Theorem 1 implies $D(\alpha_s^T \widehat{X}^{(n)}, \alpha_s^T \widehat{Y}^{(n)}) > \epsilon$ with probability $< C_1 \exp\left(-\frac{C_2}{R^4} n\epsilon^2\right)$.

Subsequently, we apply the union bound over the finite set of all grid points. The total number of points under consideration is the covering number of the unit-sphere which is $\left(1 + \frac{2R^2}{\epsilon}\right)^d$. Thus, the probability that $D(\alpha_s^T \widehat{X}^{(n)}, \alpha_s^T \widehat{Y}^{(n)}) < \epsilon$ simultaneously for all points in the grid is at least

$$C_1 \left(1 + \frac{2R^2}{\epsilon}\right)^d \exp\left(-\frac{C_2}{R^4} n\epsilon^2\right)$$

By construction, there must exist grid point $\alpha_0$ such that $||\widehat{\beta} - \alpha_0||_2 < \epsilon/R^2$. By Lemma 2

$$D(\widehat{\beta}^T \widehat{X}^{(n)}, \widehat{\beta}^T \widehat{Y}^{(n)}) \leqslant D(\alpha_0{}^T \widehat{X}^{(n)}, \alpha_0{}^T \widehat{Y}^{(n)}) + C\epsilon$$

thus completing the proof. $\qquad\square$

**Lemma 2.** *For $\alpha, \beta \in \mathcal{B}$ such that $||\alpha - \beta||_2 < \epsilon$, we have:*

$$|D(\alpha^T \widehat{X}^{(n)}, \alpha^T \widehat{Y}^{(n)}) - D(\beta^T \widehat{X}^{(n)}, \beta^T \widehat{Y}^{(n)})| \leqslant C\epsilon R^2 \tag{1}$$

*Proof.* We assume that the $\alpha$-projected divergence is larger than the $\beta$-projected divergence, and write:

$$D(\beta^T \widehat{X}^{(n)}, \beta^T \widehat{Y}^{(n)}) = \min_{M \in \mathcal{M},} \sum_{i=1}^n \sum_{j=1}^m (\beta^T x^{(i)} - \beta^T y^{(j)})^2 M_{ij}$$

recalling that $\mathcal{M}$ is the set of matching matrices defined in the main text. Let $M(\beta)$ denote the matrix which is used in the computation of the $\beta$-projected empirical Wasserstein distance (the minimizer of the righthand side of the above expression). Thus, we can express (1) as:

$$\sum_{i=1}^n \sum_{j=1}^m (\alpha^T x^{(i)} - \alpha^T y^{(j)})^2 M(\alpha)_{ij} - \sum_{i=1}^n \sum_{j=1}^m (\beta^T x^{(i)} - \beta^T y^{(j)})^2 M(\beta)_{ij}$$

$$\leqslant \sum_{i=1}^n \sum_{j=1}^m (\alpha^T x^{(i)} - \alpha^T y^{(j)})^2 M(\beta)_{ij} - \sum_{i=1}^n \sum_{j=1}^m (\beta^T x^{(i)} - \beta^T y^{(j)})^2 M(\beta)_{ij}$$

$$\leqslant \sum_{i=1}^n \sum_{j=1}^m \left[(\alpha^T (x^{(i)} - y^{(j)}))^2 - (\beta^T (x^{(i)} - y^{(j)}))^2\right] M(\beta)_{ij}$$

$$= \sum_{i=1}^n \sum_{j=1}^m \left[(\alpha - \beta)^T (x^{(i)} - y^{(j)}) \cdot (\alpha + \beta)^T (x^{(i)} - y^{(j)})\right] M(\beta)_{ij}$$

$$\leqslant \sum_{i=1}^n \sum_{j=1}^m C\epsilon R^2 M(\beta)_{ij} = C\epsilon R^2$$

$\qquad\square$

### 3.3 Proof of Theorem 3

*Proof.* Our proof relies primarily on a quantitative form of the Cramer-Wold result presented in [26]. We adapt Theorem 3.1 [26] in its contrapositive form: If $\exists\, a \geqslant 0$ such that $T_a(X,Y) > h(g(\Delta))$, then $\exists\, \beta \in \mathcal{B}$ such that

$$\sup_{z\in\mathbb{R}} \left| \Pr\left(\beta^T X \leqslant z\right) - \Pr\left(\beta^T Y \leqslant z\right) \right| > g(C\Delta) \tag{2}$$

Subsequently we leverage a number of well-characterized relationships between different probability metrics (cf. [27]) to lower bound the projected (squared) Wasserstein distance (of the underlying random variables).

Letting $K_\beta$ denote the projected Kolmogorov distance in (2), we have that the $\beta$-projected Lévy-distance, $L_\beta$ satisfies:

$$K_\beta \leqslant (1+\Phi)L_\beta \quad \text{where } \Phi := \sup_{\alpha\in\mathcal{B}} \left\{ \sup_y |f_{\alpha^T Y}(y)| \right\} \tag{3}$$

and $f_{\alpha^T Y}(y)$ is the density of the projection of $Y$ in the $\alpha$ direction.

In turn the projected Lévy $L_\beta$ is bounded above by the Prokhorov metric which itself is bounded above by the square root of the $\beta$-projected Wasserstein distance. Following the chain of inequalities, we obtain: $D(\beta^T X, \beta^T Y) \geqslant C\Delta$, to which we can apply Theorem 1 to obtain the desired probabilistic bound on the empirical projected divergence. $\qquad\square$

### 3.4 Proof of Theorem 4

*Proof.* Theorem 2 implies that with high probability, any $\beta_{S^C} \in \mathbb{R}^{d-k}$ has $D(\beta_{S^C}^T \widehat{X}_{S^C}^{(n)}, \beta_{S^C}^T \widehat{Y}_{S^C}^{(n)}) < \epsilon$, while Theorem 3 specifies the probability that there exists $\beta_S \in \mathbb{R}^k$ such that $D(\widehat{\beta}_S^T \widehat{X}_S^{(n)}, \widehat{\beta}_S^T \widehat{Y}_S^{(n)}) > d \cdot \epsilon$.

A bound for the probability that both theorems' conclusions hold may be obtained by the union bound. When this is the case, it is clear that the optimal $k$-sparse $\widehat{\beta} \in \mathbb{R}^d$ must obey the sparsity pattern specified in the statement of Theorem 4. To see this, consider any $\beta \in \mathcal{B}$ with $\beta_j \neq 0$ for some $j \in S^C$ and note that it is always possible to produce a strictly superior projection by setting $\beta_j = 0$ and distributing the additional weight $|\beta_j|$ among the features in $S$ in an optimal manner. $\qquad\square$

## 4 Derivation of semidefinite relaxation properties

Here, we provide some intuitive arguments for the conclusions in §4.3, regarding some conditions under which our semidefinite relaxation is nearly tight.

Condition (i) derives from the fact that (5) has rank one solution when the objective is approximately linear in $B$.

(ii) and (iii) are derived by noting that (5) is the Wasserstein distance between random variables $B^{1/2}X$ and $B^{1/2}Y$ where $AX$ follows a $N(A\mu_X, A\Sigma_X A^T)$ distribution when $X$ is Gaussian. Furthermore, the Wasserstein distance between (multivariate) Gaussian distributions can be analytically written as

$$W(X,Y) = ||\mu_X - \mu_Y||_2^2 + ||\Sigma_X^{1/2} - \Sigma_Y^{1/2}||_F^2$$

## Supplementary References

[1] Levina E, Bickel P (2001) The Earth Mover's distance is the Mallows distance: some insights from statistics. *ICCV* 2: 251–256.

[2] Rosenbaum PR (2005) An exact distribution-free test comparing two multivariate distributions based on adjacency. *Journal of the Royal Statistical Society Series B* 67: 515–530.

[3] Gretton A, Borgwardt KM, Rasch MJ, Scholkopf B, Smola A (2012) A Kernel Two-Sample Test. *The Journal of Machine Learning Research* 13: 723–773.

[4] Szekely G, Rizzo M (2004) Testing for equal distributions in high dimension. *InterStat* 5.

[5] Wei S, Lee C, Wichers L, Marron JS (2015) Direction-Projection-Permutation for High Dimensional Hypothesis Tests. *Journal of Computational and Graphical Statistics* .

[6] Zeisel A, Munoz-Manchado AB, Codeluppi S, Lonnerberg P, La Manno G, et al. (2015) Cell types in the mouse cortex and hippocampus revealed by single-cell RNA-seq. *Science* 347: 1138–1142.

[7] Trapnell C, Cacchiarelli D, Grimsby J, Pokharel P, Li S, et al. (2014) The dynamics and regulators of cell fate decisions are revealed by pseudotemporal ordering of single cells. *Nature Biotechnology* 32: 381–386.

[8] Ritchie M, Phipson B, Wu D, Hu Y, Law CW, et al. (2015) limma powers differential expression analyses for RNA-sequencing and microarray studies. *Nucleic Acids Research* 43: e47.

[9] Lesage S, Brice A (2009) Parkinson's disease: from monogenic forms to genetic susceptibility factors. *Human Molecular Genetics* 18: R48–R59.

[10] Linnertz C, Lutz MW, Ervin JF, Allen J, Miller NR, et al. (2014) The genetic contributions of SNCA and LRRK2 genes to Lewy Body pathology in Alzheimer's disease. *Human molecular genetics* 23: 4814–4821.

[11] Jasnow AM, Ressler KJ, Hammack SE, Chhatwal JP, Rainnie DG (2009) Distinct subtypes of cholecystokinin (CCK)-containing interneurons of the basolateral amygdala identified using a CCK promoter-specific lentivirus. *Journal of neurophysiology* 101: 1494–1506.

[12] Bormuth I, Yan K, Yonemasu T, Gummert M, Zhang M, et al. (2013) Neuronal Basic HelixLoopHelix Proteins Neurod2/6 Regulate Cortical Commissure Formation before Midline Interactions. *Journal of Neuroscience* 33: 641–651.

[13] Oh DH, Park YC, Kim SH (2010) Increased glycogen synthase kinase-3beta mRNA level in the hippocampus of patients with major depression: a study using the stanley neuropathology consortium integrative database. *Psychiatry investigation* 7: 202–207.

[14] Wu JQ, Wang X, Beveridge NJ, Tooney PA, Scott RJ, et al. (2012) Transcriptome Sequencing Revealed Significant Alteration of Cortical Promoter Usage and Splicing in Schizophrenia. *PLoS ONE* 7: e36351.

[15] Molyneaux BJ, Arlotta P, Menezes JRL, Macklis JD (2007) Neuronal subtype specification in the cerebral cortex. *Nat Rev Neurosci* 8: 427–437.

[16] Hagemann TL, Jobe EM, Messing A (2012) Genetic Ablation of Nrf2/Antioxidant Response Pathway in Alexander Disease Mice Reduces Hippocampal Gliosis but Does Not Impact Survival. *PLoS ONE* 7: e37304.

[17] Shiota J, Ishikawa M, Sakagami H, Tsuda M, Baraban JM, et al. (2006) Developmental expression of the SRF co-activator MAL in brain: role in regulating dendritic morphology. *Journal of Neurochemistry* 98: 1778–1788.

[18] Zhang L, Ju X, Cheng Y, Guo X, Wen T (2011) Identifying Tmem59 related gene regulatory network of mouse neural stem cell from a compendium of expression profiles. *BMC Systems Biology* 5: 152.

[19] Goodfellow S, Rebello M, Toska E, Zeef L, Rudd S, et al. (2011) WT1 and its transcriptional cofactor BASP1 redirect the differentiation pathway of an established blood cell line. *Biochemical Journal* 435: 113–125.

[20] Lederer CW, Torrisi A, Pantelidou M, Santama N, Cavallaro S (2007) Pathways and genes differentially expressed in the motor cortex of patients with sporadic amyotrophic lateral sclerosis. *BMC genomics* 8: 26.

[21] Jara JH, Genç B, Cox GA, Bohn MC, Roos RP, et al. (2015) Corticospinal Motor Neurons Are Susceptible to Increased ER Stress and Display Profound Degeneration in the Absence of UCHL1 Function. *Cerebral Cortex* .

[22] Li X, Yu K, Zhang Z, Sun W, Yang Z, et al. (2014) Cholecystokinin from the entorhinal cortex enables neural plasticity in the auditory cortex. *Cell Res* 24: 307–330.

[23] Gallopin T, Geoffroy H, Rossier J, Lambolez B (2006) Cortical sources of CRF, NKB, and CCK and their effects on pyramidal cells in the neocortex. *Cerebral cortex (New York, NY : 1991)* 16: 1440–1452.

[24] van der Vaart AW, Wellner JA (1996) Weak Convergence and Empirical Processes. Springer.

[25] Cramer H, Wold H (1936) Some Theorems on Distribution Functions. *Journal of the London Mathematical Society* 11: 290–294.

[26] Jirak M (2011) On the maximum of covariance estimators. *Journal of Multivariate Analysis* 102: 1032–1046.

[27] Gibbs AL, Su FE (2002) On Choosing and Bounding Probability Metrics. *International Statistical Review* 70: 419–435.

## Footnotes

[1]available in NCBI's Gene Expression Omnibus (under accession GSE60361)