[Reviews · NeurIPS 2015]

Submitted by Assigned_Reviewer_1

The paper give a rather original approach to characterizing differences between two distributions (e.g. populations) by linear projections. Experiments on synthetic and real data seem to support the conclusions. Looks like a very solid and original paper.

Unfortunately I have low confidence for this review mainly due to my ignorance of relevant literature.
Summary: Interesting and algorithmically sophisticated paper on a method trying to find differences between populations, going beyond mere classification.

Submitted by Assigned_Reviewer_2

The authors proposed a new dimensionality reduction method that finds the most different direction between input X and Y. The novelty of the proposed method is to use the squared Wasserstein distance as discrepancy measure and can be solved by semidefinite programming. Through experiments, authors showed that the proposed method compares favorably with existing methods.

Quality: The method technically sounds.

Clarity: The paper is well written and easy to follow.

Originality: The approach is new. The problem in this paper is similar to the one in Transfer component analysis, which finds a subspace that have small discrepancy between two datasets.

Significance:

The formulation with Wasserstein distance + SDP is interesting. Thus, it would have some impact in ML community.

Detailed comments: 1. The problem can be easily formulated by using simple Lasso. For example, if we add positive pseudo labels for X and negative pseudo labels for Y and solve the problem ||Y - Z^t \beta||_2^2 + \lamda ||beta||_1, you may be able to obtain similar results. Actually, this approach only useful if X and Y are linearly related, thus it can be a good baseline.

2. Is it possible to extend the algorithm to nonlinear case? 3. For the problem, transfer component analysis can be used to find a most different direction. (Although TCA was originally proposed for finding common subspace, it can be easily applied for your task). http://www.cse.ust.hk/~qyang/Docs/2009/TCA.pdf

Summary: The proposed formulation is interesting. If author add can add simple Lasso based baseline, it would be a plus.

Submitted by Assigned_Reviewer_3

The proposed approach is motivated by the "Cramer-Wold device", which ensures the existence of a linear projection that differentiates two distributions. The authors apply the Wasserstein metric directly on samples from both distributions, and show favorable theoretical properties of such an approach under reasonable assumptions (such as bounded domain variables). The analysis is extended to distributions that differ only in a few dimensions, by considering sparse projections. The resulting computational problem turns out to be non-convex. The authors propose a semidefinite program relaxation and further pre-processing to estimate a solution. Empirical evaluation is provided on simulated and real datasets, including an application to bioinformatics.

The paper is well written and clearly lays out the main ideas. The paper dresses a problem of interest to the community using a novel technique and sufficient theoretical and empirical validation.

Minor issues: - I would encourage the authors to include more description and legends in each figure so they are more self contained and simplify interpretation. - the term "Cramer-Wold device" seems to be fairly non-standard outside of the reference [10]. Would the authors prefer the more standard "Cramer-Wold theorem"?

Suggestions for future work: The presented theoretical results are concerned with the sample complexity as compared to the optimal linear projection. It would also be useful to characterize the relationship between the induced divergences and more standard divergences computed directly between the high dimensional distributions, as this would help clarify the convergence in distribution implied by the Cramer Wold Theorem.
Summary: The authors propose a technique to differentiate between distributions, based on the maximal univariate Wasserstein divergence between projections of the underlying random variables. The paper dresses a problem of interest to the community using a novel technique and sufficient theoretical and empirical validation.

Submitted by Assigned_Reviewer_4

The paper proposes to find sparse projection bases that maximise the Earth Mover's Distance (EMD) between distributions. To the

best of my knowledge component analysis using EMD has received limited attention (except [A] for NMF). A non-convex optimisation problem is formulated and solved via a kind of alternating optimization. Some experiments (which importance I cannot fully evaluate) show quite some improvement over the state-of-the-art.

[A] Sandler, Roman, and Michael Lindenbaum. "Nonnegative matrix factorization with earth mover's distance metric for image analysis." Pattern Analysis and Machine Intelligence, IEEE Transactions on 33.8 (2011): 1590-1602.

After reading the rebuttal I believe the paper is of NIPS quality and recommend acceptance.
Summary: The paper proposes to find sparse projection bases that maximise the Earth Mover's Distance (EMD) between distributions. To the

best of my knowledge component analysis using EMD has received limited attention. An alternating optimisation procedure is proposed to find the bases. Some experiments (which importance I cannot fully evaluate) show quite some improvement over the state-of-the-art. I recommend acceptance.

Author Feedback
Author rebuttal: We thank the reviewers for their comments and interest. In our response, we abbreviate reviewer names (eg. R1 = Assigned_Reviewer_1).

R2 proposes a baseline method to compare with. Our interpretation of the comment is that in the expression ||Y - Z^t beta||_2, R2 uses Z to denote the feature-vector and Y a 0-1 label, so this proposal corresponds to standard least-squares regression (with lasso). Generally, logistic (lasso) regression is preferable for binary responses [1]. As we already evaluated our approach against the latter method (Figure 1b), the proposed comparison seems unnecessary given the space constraints.

R2 suggests extending our method to nonlinear projections. Due to the Cramer-Wold device, our linear PDA can already capture general effects as the marginal distribution after projection remains unconstrained. While nonlinear extensions are certainly promising, we believe this would entail a separate method (better introduced in another paper) that leverages ideas from RKHS/kernel theory, principal curves, or manifold learning. Here, the primary challenge would be retaining interpretability.

R2 suggests comparing with a max-rather-than-min variant of Transfer Component Analysis [2]. This is certainly a related method, but the nonlinearity of the feature mapping hinders interpretability (and if feature maps are not sufficiently complex/nonlinear, i.e. corresponding to a "characteristic" kernel, then MMD - the divergence optimized in TCA - is not a proper distance between distributions because the mean map is no longer injective [3]). Without additional modification to enhance interpretability, this TCA-variant is better suited for two-sample testing. We have already compared our approach against the closely related MMD statistic (intended for high-dimensional problems) [3] and many other popular two-sample tests. Furthermore as R2 states, the actual TCA method is not relevant and would require modification.

R6 suggests comparing with the MMD two-sample test from A. Gretton [3]. We did this in Figure 1c (green curve), but there may be confusion because we poorly refer to this method as "Mean Map Discrepancy" (referencing an earlier name) rather than "Maximum Mean Discrepancy". We will change to the proper latter name. Furthermore, our methods have additional uses beyond two-sample testing / measuring distance between distributions, unlike MMD (the weights in our projection have interpretations which provide additional characterization of the inter-population differences beyond a single p-value or computed distance).

R3 brings up a good point that Earth Mover distances have received limited attention in component analysis. We shall mention this as background and include the reference provided by the reviewer, along with highlighting other works that successfully employ Wasserstein metrics surveyed in [4].

This should also address R6's concern why Wasserstein divergence is of particular interest. We can also further motivate this metric intuitively. For example, the Wasserstein distance provides a dissimilarity measure between populations that integrates both the fraction of individuals which are different together with the magnitude of the differences.

There is an interesting interpretation of Wasserstein PDA (which we omitted due to space, but could include, e.g. in supplement):
Consider a two-class problem with paired samples (x_i, y_i) where X, Y follow two multivariate Gaussian distributions. The first principal component of the paired-differences X - Y (in PCA without centering) equals the direction which maximizes the projected Wasserstein distance between the distribution of X - Y and a delta distribution at the origin. Thus (uncentered) PCA can be viewed as a special case of PDA in the Gaussian, paired-sample-differences case.

R4 provides a nice Cramer-Wold reference that we will include in our final draft.

R5 suggests more detailed Figure 1 legend/captions which we will address.

R5 provides ideas for future theoretical investigation, particularly the desire for connections to more-standard divergences computed between the original high-dimensional distributions. We agree this is a good idea.

R5 views term "Cramer Wold Theorem" as more standard than "Cramer Wold device". We believe the latter is actually more common (cf. [5], a standard text) but are happy to use whichever is more understandable.

[1] Brannick MT. Logistic Regression.
http://faculty.cas.usf.edu/mbrannick/regression/Logistic.html

[2] Pan SJ, Tsang IW, Kwok JT, Yang Q (2009) Domain Adaptation via Transfer Component Analysis. IJCAI.

[3] Gretton A, Borgwardt KM, Rasch MJ, Scholkopf B, Smola A (2012) A Kernel Two-Sample Test. JMLR 13: 723-773.

[4] Levina E, Bickel P (2001) The Earth Mover's distance is the Mallows distance: some insights from statistics. ICCV 2: 251-256.

[5] Durrett, R. (2010) Probability Theory and Examples. 4th edition: pp 176.